# Erythropoiesis and Malaria, a Multifaceted Interplay

**DOI:** 10.3390/ijms232112762

**Published:** 2022-10-23

**Authors:** Aurélie Dumarchey, Catherine Lavazec, Frédérique Verdier

**Affiliations:** 1Inserm U1016, CNRS UMR8104, Université Paris Cité, Institut Cochin, 75014 Paris, France; 2Laboratoire d’Excellence GR-Ex, 75015 Paris, France

**Keywords:** ineffective erythropoiesis, dyserythropoiesis, malarial anemia, *Plasmodium*

## Abstract

One of the major pathophysiologies of malaria is the development of anemia. Although hemolysis and splenic clearance are well described as causes of malarial anemia, abnormal erythropoiesis has been observed in malaria patients and may contribute significantly to anemia. The interaction between inadequate erythropoiesis and *Plasmodium* parasite infection, which partly occurs in the bone marrow, has been poorly investigated to date. However, recent findings may provide new insights. This review outlines clinical and experimental studies describing different aspects of ineffective erythropoiesis and dyserythropoiesis observed in malaria patients and in animal or in vitro models. We also highlight the various human and parasite factors leading to erythropoiesis disorders and discuss the impact that *Plasmodium* parasites may have on the suppression of erythropoiesis.

## 1. Introduction

Malaria is one of the most important human infectious diseases and particularly affects populations living in tropical and subtropical countries. Nowadays, although some antimalarial drugs are available, malaria remains a major public health problem with 241 million cases and 627,000 deaths per year [1]. This infection is caused by the protozoan parasite *Plasmodium*. Five species are responsible for malaria in human beings: *P. vivax*, *P. malariae*, *P. ovale*, *P. knowlesi* or *P. falciparum*. While *P. vivax* is the most widespread, *P. falciparum* is responsible for almost the totality of severe and lethal malaria cases. Other species can also infect vertebrates including mice, such as *P. berghei*, *P. chabaudi*, *P. vinckei* and *P. yoelii*. Malaria clinical symptoms result largely from the replication of asexual parasite stages. Signs of infection are mainly fever, flu-like symptoms associated with a mild to profound anemia, which can lead to serious complications and even death in the case of severe malaria [2].

At the end of the 19th century, Laveran detected for the first time the parasite responsible for malaria in human blood [3]. Then, the presence of the parasite was confirmed in patient autopsies, showing parasites sequestered in deep organs [4]. Several studies have explored the location of parasites through human tissues. Different parasite reservoirs were first discovered by microscopy and later by molecular analysis, among them the brain, liver, the spleen and the bone marrow (BM) [5,6,7,8]. The presence of *Plasmodium* parasites in the BM raises questions about their impact on erythropoiesis, which is the process leading to the production of red blood cells (RBCs) from hematopoietic stem cells (HSCs) in the BM. Erythropoiesis is a very dynamic process able to generate 3 × 10^6^ RBCs per second, taking place in the erythroblastic island. This specialised niche in the BM is formed by a nursing macrophage surrounded by erythroblasts at different stages of differentiation. HSCs first generate common myeloid and lymphoid progenitors and then give rise to lineage-restricted progenitors. The MEP (Megakaryocyte Erythroid Progenitor) differentiate into erythroid progenitors, BFU-E (Burst Forming Unit–Erythroid) and then CFU-E (Colony Forming Unit–Erythroid). Terminal adult erythropoiesis begins with the differentiation of CFU-E in proerythroblasts (Pro-E), which then are differentiated into early basophilic-1 erythroblasts (Baso-E1), late basophilic erythroblasts (Baso-E2), polychromatophilic erythroblasts (Poly-E) and orthochromatic erythroblasts (Ortho-E). Finally, Ortho-E enucleate and generate reticulocytes and pyrenocytes. Reticulocytes mature first in the BM and then are directed into the blood circulation where they finally become discoid red cells [9,10].

Although malarial anemia is thought to be largely caused by hemolysis and splenic clearance of infected and uninfected erythrocytes [11,12,13,14,15], abnormal erythropoiesis has been observed in malaria patients and could potentially be instrumental in anemia [16]. The relation between erythropoietic defects and malarial anemia has been poorly investigated so far; however, recent findings may provide new insights.

This review summarises the erythropoiesis abnormalities observed in malaria, either in vivo in malaria-infected patients and in murine malaria models, or ex vivo in hematopoietic stem and progenitor cells (HSPCs) and erythroid cell lines. We review our current knowledge of the causes of erythropoietic disorders and discuss how *Plasmodium* infection may impact this process.

### Plasmodium Parasite Sequestration in the Bone Marrow

*Plasmodium* has a complex life cycle with asexual and sexual stages that develop through various tissues in intermediate and definitive hosts (Figure 1). First, sporozoites are transmitted from female *Anopheles* bites to human beings. Sporozoites migrate into hepatocytes where they multiply for several days, leading to numerous merozoites that are released into the bloodstream. Then, merozoites invade RBCs, where the developmental asexual cycle occurs: parasites mature from rings to trophozoites and become schizonts within 48 h. RBCs infected with mature asexual parasites (trophozoites and schizonts) adhere to endothelial cells, causing their sequestration in capillaries and venules of several organs [17,18,19]. In contrast, RBCs containing ring-stage parasites are not retained in vasculature and circulate in the bloodstream. During each round of asexual replication, a sub-population of parasites differentiates into sexual stages, called gametocytes, which are responsible for transmission from human beings to mosquitoes. Gametocyte maturation takes about ten days and is classically divided into five morphological stages (I–V). Immature gametocytes are absent from the bloodstream and are sequestered in deep tissues whereas only mature gametocytes (stage V) appear in the peripheral blood where they are accessible for mosquito bites. Then, parasite sporogonic development continues in the mosquito [20].

Several case studies analysed the distribution of asexual and sexual parasites in different organs. Microscopy analysis of BM and blood from Gambian children showed that *P. falciparum* asexual and mature gametocytes (stage V) were equally present in the BM and the blood, whereas immature gametocytes (stage I–IV) were preferably localised in the BM [6]. Although this latter study solely relied on parasite morphology, accumulation of *P. falciparum* in the BM was recently confirmed by immunohistochemical or immunofluorescent labelling approaches and quantified by qRT-PCR [7,8,21]. As observed for *P. falciparum*, *P. vivax* immature gametocytes are enriched in BM aspirations in comparison to peripheral blood but to a lesser extent [22,23]. The total asexual parasitemia seems identical in peripheral blood compared to BM, although an enrichment of ring stages and immature gametocytes is observed. The presence of ring stages in the BM is probably due to the strict preference of *P. vivax* for invading young reticulocytes expressing high levels of Cluster of Differentiation 71 (CD71) that are essentially present in this compartment [24]. Recent studies from splenectomised patients suggest that the main reservoir for *P. vivax* is probably not the BM but rather the spleen, and further investigations will be necessary to confirm this interesting data [25,26]. *Plasmodium* asexual parasite and gametocyte enrichment in the BM and the spleen was confirmed in distinct vertebrate species: in mice infected by *P. berghei* [27] as well as in a humanised mouse model infected by *P. falciparum* [28], and in non-human infected primates where accumulation of *P. vivax* parasites was observed in the BM [29].

Importantly, histological analyses of ex vivo and autopsy specimens from *P. falciparum* malaria-infected patients revealed the presence of immature gametocytes in the BM extravascular space [7,30]. More precisely, parasites are located near to the erythroid precursor cells in contact with the erythroblastic islands [7]. These observations suggest that parasites may sequester in this niche either by adhering to the erythroblasts or by infecting them. The hypothesis of cell–cell adhesion is not favoured since gametocyte-infected erythrocytes do not adhere to erythroblasts [31] and several studies clearly demonstrated the ability of *Plasmodium* to invade erythroid precursors. Indeed, *P. vivax* and *P. falciparum* asexual stages were observed in vitro in erythroblasts derived from CD34+ HSPCs. Asexual parasites were detected at the polychromatic, orthochromatic and reticulocyte stages, in which the entire asexual cycle can be achieved [32,33,34,35]. Furthermore, a recent study suggested that the erythroblast could serve as a host cell for *P. falciparum* gametocytes [21]. In this study, in vitro infections of human primary erythroblasts derived from granulocyte colony-stimulating factor–mobilised peripheral blood or from BM aspirate revealed that gametocytes could fully develop in human erythroblasts from the Poly-E, confirming previous evocative observations [34]. This discovery was supported by in vivo analyses of BM smears from a malaria-infected patient with a gametocyte marker, anti-Pf11.1 antibody [36], showing the presence of gametocytes inside nucleated erythroid cells [21]. In line with these observations, recent studies in mice reported that *P. berghei* could also invade murine erythroblasts [37,38]. These results suggest that the infection of erythroblasts likely contributes to *Plasmodium* parasite sequestration in the BM, in addition to infected-RBC adhesion to BM mesenchymal stem cells [39] and mechanical retention due to the important stiffness of the infected RBCs (iRBCs) [40,41,42]. The erythroid niche in the BM may provide a unique environment where the parasite finds appropriated metabolites and concentration of oxygen.

## 2. Ineffective Erythropoiesis and Dyserythropoiesis in Malaria

Ineffective erythropoiesis is described as an abnormal proportion of erythroblasts in the BM leading to anemia. During steady-state human erythropoiesis, the normal ratio between erythroblasts is of 1 for Pro-E; 2 for Baso-E1; 4 for Baso-E2; 8 for Poly-E and 16 for Ortho-E. In the case of ineffective erythropoiesis, these ratios are unbalanced, leading to a decreased number of red cells. In contrast, dyserythropoiesis refers to qualitative abnormality of erythropoiesis with morphological defects of erythroblasts, which may finally be responsible for a decreased erythrocytes production. Ineffective erythropoiesis associated to dyserythropoiesis are general features found in malaria in several *Plasmodium* species. All stages of erythropoiesis are impacted by *Plasmodium* infection. A drastic decrease in progenitors (BFU-E and CFU-E) has been observed in murine BM after infection by *P. berghei* [43] and in patients infected by *P. falciparum* [44]. Other studies described impaired erythropoietic response to anemia with a low rate of red cell precursors and RBCs in human malaria patients [45,46,47] and these data were later confirmed in murine BM [48]. More recently, molecular studies in patients infected with *P. vivax* showed an abnormal ratio of erythroblasts [22,23].

Low reticulocytosis is also a striking mark of ineffective erythropoiesis in patients. The low and inadequate percentage of reticulocytes was first described in patients with *P. falciparum* and *P. vivax* infections [45,49,50] and was confirmed in mice infected with *P. berghei*, *P. chabaudi* or *P. yoelii* [43,48,51,52,53], as well as in monkeys infected by *P. vivax* [54]. The low reticulocyte production was also detected in in vitro culture of *P. falciparum*-infected human primary erythroblasts derived from HSPCs [21].

Ineffective erythropoiesis is also associated with an inhibition of the erythroid cell proliferation and cellular division [47,53,55]. A deregulation of the cell cycle was first observed in children infected with *P. falciparum*, in which a decreased number of cells in G1 and a decreased S/G2 ratio occurred in Baso-E. This latter observation was also reported in Poly-E, using 3H-thymidine autoradiography, suggesting an accumulation in the G2 phase [56]. Conversely, a prolonged S phase was reported in murine erythroid precursors using propidium iodide incorporation [48]. Further investigations will be necessary to decipher more precisely the cell cycle for each stage of erythroid differentiation in malaria patients.

Erythropoietin (EPO) is the key regulator of erythropoiesis, sustaining the proliferation and survival of late erythroid progenitors and precursors. In the case of low RBC count, the resulting hypoxia causes an increase in serum EPO levels leading to the expansion of the erythroid progenitors and precursors, to reticulocytosis and eventually a restored RBC count. Interestingly, the inhibition of BM erythropoiesis in patients infected by *P. falciparum* [57] or in mice infected with rodent malaria parasites [53,58] occurs despite increased levels of EPO in blood, suggesting that *Plasmodium* infection causes suppression of the BM response to EPO.

Besides disrupting the efficiency of erythropoiesis, malaria parasites are also associated with dyserythropoietic features in BM. Light and electron microscopic observations of BM aspirates from patients infected with *P. falciparum* revealed erythroid hyperplasia and ultrastructural nuclear abnormalities such as multinuclearity, nuclear fragmentation, internuclear bridges and irregular nuclear shapes [8,47,56,59,60,61]. Infection with *P. vivax* also generates dyserythropoiesis as described in adult BM aspirates from patients with abnormal nucleus (multiple nuclei, budding nuclei, …) as well as intercytoplasmic bridge [22,23,50].

Some molecular studies highlighted the significant erythropoietic suppression contributing to malarial anemia. Microarray and RT-qPCR analyses of infected murine spleen and BM, the two primary sites of erythropoiesis, revealed a downregulation of erythroid-specific transcripts. Among them, a decrease in major transcription factors (Gata-1, Nfe2, Eklf or Gfi1b), erythroid-specific markers (Gpa, Band3) and transcripts involved in heme biosynthesis pathways confirmed suppressed erythropoiesis [62,63]. More recently, transcriptional changes were also observed in BM from *P. vivax*-infected patient aspirates, using RNA sequencing and RT-qPCR. This study reported a decrease in transcripts of genes involved in erythropoiesis (TAL1, GATA1, NFE2, ARID3A, ALAS1 and ALAS2) while those related to cytokine secretion or complement cascade were upregulated [23]. A microarray profiling performed during in vitro co-culture of *P. falciparum* parasites and Ortho-E showed that many erythroid genes were upregulated, a large number of them encoded mediators of metabolism, NRF2-mediated oxidative response, cellular stress response or mitochondrial dysfunctional pathways. These data suggest that upon infection, erythroblasts may respond and adapt to parasite factors by modifying gene expression [64].

## 3. Human and Parasite Factors Affecting Erythropoiesis

### 3.1. Extracellular Vesicles

Extracellular vesicles (EVs) are small membrane-bound vesicles involved in cellular communication, physiological processes, and immune regulation in many species [65,66]. EVs can be secreted by various cell types; interestingly, it has been described that in the BM, the most abundant EVs originate from erythroid cells [67]. EVs are classified in different groups depending on their cellular origin, size and biological functions. Apoptotic bodies, the biggest ones with 5000 nm, are secreted from the cell surface during apoptosis; microvesicles, between 100–1000 nm, are released by budding or shedding from the plasma membrane; and exosomes, the smallest vesicles (range from 30 to 150 nm), are derived from the endosomal system [68,69,70].

EVs are present in healthy individuals, but their secretion is enhanced under pathological conditions, such as parasite infection [71]. In the case of *Plasmodium* infection, a major increase in EVs derived from iRBCs (iEVs) in the plasma of patients is reported, correlating with disease severity [72,73,74]. According to the analytical approaches recommended in the guidelines of vesicle identification [75], EVs released during malaria infection are exosomes [76,77]. EVs released from *Plasmodium*-iRBCs are then internalised by other cells: endothelial cells, spleen fibroblasts, immune cells, or erythroid cells. EVs allow an exchange of material that modify the biological functions of these cells, by altering the vascular functions, facilitating cytoadherence, activating the immune cells, modulating mechanical properties of RBCs or regulating erythropoiesis [21,78,79,80,81]. When EVs are incubated in vitro with Ortho-E, they induce a delay of enucleation and an increase in oxidative stress [21]. In *P. falciparum*-infected erythroblasts, infection increases the production of EVs by erythroblasts to a similar level as by erythrocytes, inducing as well a reduction in reticulocytes production by bystander erythroblasts [21].

The content carried by EVs depends on the parasitic stage [82] and on in vitro growing density of the parasite [83]. Recently, Abou Karam et al. demonstrated that two EVs subpopulations of distinct size are secreted by *P. falciparum*-iRBCs, differing in their protein content and specific membrane composition. Their results suggest that these EVs may have the ability to fuse to distinct target cells [84]. EVs from *Plasmodium*-iRBCs contain proteins, lipids, metabolites, and nucleic acids, originating from both the host and the parasite. The protein composition of EVs from malarial iRBCs is well described, via proteomic approaches on EVs from *P. falciparum*-iRBCs cultivated in vitro [78,81] and from in vivo models with EVs from the plasma of *P. falciparum*-infected malaria patients [85] or from the plasma of *P. yoelii or P. berghei*-infected mice [76,77,86]. These studies highlighted that there are far more proteins in patient EVs than in control EVs and that they have a role in metabolic processes, host immune response, structure, or adhesion of the parasite. Several studies have described that EVs represent a mechanism by which the parasite promotes immune escape and can modify the host immune responses by increasing cytokines production; Interleukin (IL)-6, IL-12, IL-1, IL-10, tumor necrosis factor α (TNFα) and interferon γ (IFNγ) [78,87,88,89]. These inflammatory cytokines could severely impact erythropoiesis, as described in Section 3.3. Cytokines.

Small RNAs contained in EVs could also contribute to erythropoiesis suppression. The small RNA composition of EVs from iRBCs during *Plasmodium* infection is well characterised, using RNA sequencing of EVs from RBCs infected with *P. falciparum* in vitro [90,91] and using RT-qPCR on EVs from *P. falciparum* or *P. vivax*-infected malaria patients [92]. Different categories of small regulatory RNAs were found in iEVs originating either from the host or the parasite: miRNAs, tRNA, Y-RNA, vault-RNA, SRP-RNA, U-RNA, Piwi-RNA and sno-RNA. A high level of host miRNAs was observed, which regulate the expression of several genes at the post-transcriptional level. The Ago2-miRNAs complex has also been found in iEVs allowing the formation of the RISC complex, which can silence parasite and host target genes [79].

All miRNAs detected in EVs from iRBCs originate from the host cell since *Plasmodium* parasites have no RNAi machinery and cannot produce their own miRNAs [93,94]. Interestingly, the main miRNAs found in EVs produced by iRBCs are the same as those in uninfected RBCs, but at different concentrations [91,95]. These miRNAs (miR-451a, miR-486-5p, miR-144-3p or miR-92a-3p) are known to have an important impact on erythropoiesis, as a regulator of erythroid cell differentiation [96,97,98,99,100]. They are also associated with oxidative stress and severe anemia [101,102]. Accordingly, the release of miRNAs from *Plasmodium*-infected RBCs can modify erythroblasts gene expression and may regulate erythropoiesis [93,103]. Overall, EVs are playing an important role in malarial anemia by manipulating the erythroid host. EVs act either directly on erythroblasts or indirectly essentially through the inflammatory response mediated by immune cells and the secretion of cytokines (Figure 2).

### 3.2. Hemozoin

Hemozoin (Hz), a malaria pigment crystal, is a scaffold of heme dimers and polyunsaturated fatty acids [104]. It is formed in the food vacuole of *Plasmodium* parasites during the digestion of the host RBC hemoglobin. The parasites digest up to 75% of hemoglobin, which is the main source of parasite nutrients. The digestion of hemoglobin produces free heme, which is then polymerised into Hz to avoid heme toxicity [105,106]. During the parasite life cycle, the malaria pigment is released with merozoites during the rupture of parasitised-RBC and then is phagocytised by macrophages, monocytes, neutrophils or dendritic cells [107]. In *Plasmodium* infections, high levels of Hz are found in the BM and a correlation between Hz accumulation in this compartment and the severity of the anemia is observed in patients [8,108,109] and in murine models [110].

Several in vitro experiments indicated that Hz disrupts the development of erythroid cells. Upon incubation with Hz, decreased numbers of BFU-E and CFU-E were observed in colony forming cells assay from human HSPCs [111,112]. Moreover, this ex vivo culture of erythroblasts showed that Hz delays cell cycle progression, resulting in a decreased proliferation of erythroid progenitors [111,112]. In addition, Reactive Oxygen Species (ROS) production induced by Hz in Baso and Poly-E leads to cell apoptosis mediated through the activation of caspases (caspases 3, 9 and 8) [113]. Furthermore, lipid peroxidation of Hz induces the production of 4-hydroxynonenal (4-HNE), which is a bioactive aldehyde molecule able to bind to proteins and DNA, thus forming 4-HNE adducts [114,115]. 4-HNE, like its precursor Hz, has an inhibitory effect on erythropoiesis: it induces a reduction in the number of erythroid progenitors, an accumulation of cells in the G0/G1 phase and a delay in erythroid differentiation [111,112]. Proteins associated with Hz may also contribute to the suppression of erythropoiesis [116].

In the presence of Hz, microarray analysis showed an upregulation of the transcription factors controlling red cell differentiation and apoptosis, as well as survival during cell stress, such as DNA damage-induced transcript 3 (DDIT3) and DNA damage-induced transcript 4 (DDIT4). In addition, a downregulation of genes implicated in cell cycle, nuclear mRNA splicing, and apoptosis was observed [117]. A similar study previously showed the dysregulation of genes involved in cell cycle (p53, p21) or erythroid development (GATA1, TfR1, SCFR, IL3R and EPOR) in the presence of Hz and 4-HNE [112].

In addition to the direct effect of Hz on the development of erythroid cells, the immune responses generated by Hz may also indirectly impact erythropoiesis. Hz is a major regulator of the innate immune response, stimulating the secretion of cytokines, chemokines, and peroxides by these phagocytic cells. This process has a protecting effect for the host against malaria, but may also have detrimental consequences on erythroblasts, which lead to anemia [110,118,119] (see part 3.3. cytokines). Furthermore, some reports suggested that the Hz inflammatory effects are due to the oxidative stress generated by the production of ROS via heme catabolism and the Fenton reaction. Indeed, ROS have been detected in phagocytic cells and erythroblasts after incubation with Hz [113,120,121].

### 3.3. Cytokines

After *Plasmodium* infection, the innate immune system rapidly detects the parasite and induces a strong inflammatory response. Inflammatory mediators such as cytokines, chemokines, or nitric oxide (NO) are highly produced by monocytes/macrophages from distinct organs in order to eliminate infected RBCs. These mediators act as an antimalarial host defense by mediating immune response while having a negative impact in the BM by repressing erythropoiesis. For example, TNFα and IFNγ are potent inhibitors of erythropoiesis [122]: TNFα significantly decreases human erythroid progenitor cell proliferation [123] while IFNγ inhibits the proliferation and the differentiation of erythroblasts [124]. *P. vinckei* infection induces a drastic increase in TNFα in the serum of mice and TNFα injection provokes a dyserythropoiesis in the BM and an enhanced phagocytosis of erythroblasts [125]. In *P. falciparum* malaria, a significant increase in IFNγ plasma levels was reported [126], suggesting that this cytokine may contribute to malarial anemia by inhibiting erythropoiesis. However, a *P. vivax* in vitro study using erythroblasts derived from HSPCs demonstrated that the parasite can inhibit erythroid development independently of these two cytokines [55].

The levels of Interleukins are also clearly modified during *Plasmodium* infection with consequences on erythropoiesis. Low levels of Interleukin-10 (IL-10) and an abnormal low ratio of IL-10/TFNα in patient serum are associated with the severity of malarial anemia [127,128]. Indeed, IL-10 downregulates TNFα leading to the stimulation of erythropoiesis [129] and upon *Plasmodium* infection, a reduced level of IL-10 may result in erythropoiesis repression. Interleukin-12 (IL-12) is a direct stimulator of erythropoiesis increasing the production of BFU-E and CFU-E, leading to an enhanced erythrocyte count [130]. Accordingly, mice infected with *P. chabaudi* showed a deficient production of IL-12, resulting in anemia and fatal malaria, whereas treatment with IL-12 is able to correct the anemia [131]. Furthermore, polymorphisms in the IL-12/IL-12 receptor genes are associated with protection against severe malarial anemia [132]. Moreover, serum levels of Interleukin-6 (IL-6) were elevated in patients with *P. falciparum* infection [133]. This inflammatory cytokine acts indirectly on erythroblasts and contributes to the regulation of iron homeostasis by increasing hepcidin expression in the liver. The hepcidin hormone is the major regulator of iron metabolism and an increase in hepcidin leads to a decrease in iron availability for erythropoiesis [134,135].

Plasma MIF (macrophage migration inhibitory factor), another pro-inflammatory cytokine, is also enhanced in malaria-infected patients and is associated with the severity of anemia. MIF suppresses erythroid colony formation (BFU-E, CFU-E) at concentration found in the plasma of patients and synergises with known antagonists of erythropoiesis such as TNFα and IFNγ [136,137].

Several evidences show that chemotactic cytokines, such as the chemokines Macrophage Inflammatory Proteins (MIP) MIP-1α, MIP-1 β and regulated on activation, normal T-cell expressed and secreted (RANTES) are also involved in malarial anemia by erythropoietic suppression [138,139]. Circulating levels of MIP-1α and MIP-1β are elevated in patients with mild and severe malaria, whereas those of RANTES are decreased with increasing anemia severity [118]. These chemokines may contribute to the ineffective erythropoiesis since MIP-1 has a suppressive effect on erythroid progenitors [140] and RANTES prevents their apoptosis [141].

Finally, the pro-inflammatory mediator NO inhibits the proliferation and the erythroid differentiation of human primary erythroblasts and contributes to anemia [142,143]. Several studies reported the elevation of NO in blood upon *Plasmodium* infection in children with malarial anemia [144,145]. Interestingly, recent data show that *P. falciparum* gametocytes are phagocytosed by immortalised mouse BM-derived macrophages and late gametocytes initiate the production of NO, cytokines, and chemokines [146]. Although these data have to be confirmed in a more physiological human setting, they suggest that gametocytes may stimulate the secretion of erythropoiesis-inhibiting factors by macrophages constituting the erythroblastic islands, which are the specialised niches for erythropoiesis.

## 4. Conclusions

Anemia is an important clinical manifestation of malaria due to the increased lysis of RBC and to decreased RBC production as a result of ineffective erythropoiesis and dyserythropoiesis. The mechanisms responsible for the suppression of erythropoiesis are multiple and complex and a better understanding of them may provide comprehensive insight in the pathophysiology of malaria, leading to discoveries for future treatments of anemia.

## Figures and Tables

**Figure 1 ijms-23-12762-f001:**
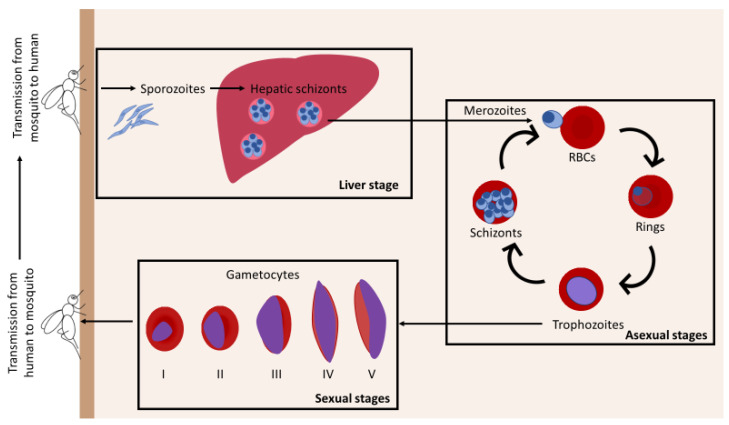
Plasmodium life cycle. During its meal, infected female *Anopheles* inject sporozoites into human beings. Sporozoites reach the liver and multiply for several days, resulting in the release of merozoites into the bloodstream. Then, merozoites invade RBCs and mature from rings to trophozoites and schizonts (asexual stages). A small portion of parasites differentiates in gametocytes (sexual stages). Gametocyte development is divided into five morphological stages (I–V), I–IV are sequestered in the BM and stage V is released into the peripheral blood, enabling the transmission to the mosquito.

**Figure 2 ijms-23-12762-f002:**
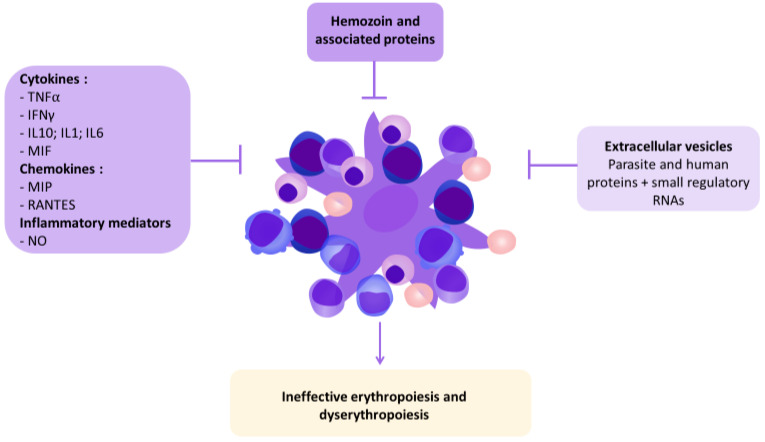
Human and parasite factors affecting erythropoiesis in malaria.

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
