# Peer review of "Erythropoiesis and Malaria, a Multifaceted Interplay"

_ijms, 2022, doi:10.3390/ijms232112762_

Round 1

Reviewer 1 Report

The authors are experts in the field of malarial parasites and their interaction with red blood cells and they have written a very nice review of the relationship between the parasite and the RBC. One of the authors (Neveu) has also recently published a review on erythroid cells and malaria parasites but there is not much overlap (I compared the two articles).

The current article is very interesting and gives use details of the interactions of the malarial parasite and the RBC and explains how malaria, by invading the RBC precursors, inhibits erythropoiesis. I had previously thought that malarial anemia was caused by two mechanisms: hemolysis and depression of RBC production by a mechanism of “chronic disease” anemia, mediated by hepcidin and other cytokines. The authors mention the hemolysis and splenic clearance of erythrocytes in lines 47-50. However I found myself wondering whether the anemia is met with a compensatory reticulocytosis. I have seen few patients with active malaria as I work in a country in which malaria is not endemic. So it may help focus the reader’s mind on the ineffective erythropoiesis possibly resulting from malarial parasites if the authors put in here (like 50) a few sentences on the fact that the malarial anemia is not accompanied by reticulocytosis (or is it??) due to…whatever factors they think are at play. Perhaps this seems very elementary however many hematologists in the western world do not see much anemia so this would be helpful in focusing on the importance of what is going on in the bone marrow.

Other than this suggestion, I do not have any suggestions for revisions. The illustrations are nicely done and the manuscript is interesting. A bit of minor corrections for the English are required for example  in a number of places the authors write “erythropoiesis abnormality” when the word “erythropoietic” should be used. There are a few incorrect uses of grammar but these are very minor.

Reference:

Neveu G, Lavazec C. Erythroid cells and malaria parasites: it's a match! Curr Opin Hematol. 2021 May 1;28(3):158-163.

Author Response

We are very grateful to reviewer 1 for his critical reading of the manuscript and his comments.

 As rightly discussed by the reviewer, malarial anemia is not accompanied by an expected reticulocytosis to compensate the decrease in red blood cells. We had already discussed this point in a paragraph (line 118 to 121): “Low reticulocytosis is also a striking mark of ineffective erythropoiesis in patient. The low and inadequate percentage of reticulocytes was first described in patients with P. falciparum and P. vivax infections [45, 49, 50] and was confirmed in mice infected with P. berghei, P. chabaudi or P. yoelii [43, 48, 51–53], as well as in monkeys infected by P. vivax [54]. The low reticulocyte production was also detected in in vitro culture of P. falciparum-infected human primary erythroblasts derived from HSPCs [21].”

We replace the word erythropoiesis abnormalities by erythropoietic abnormalities as suggested or abnormal erythropoiesis.

Reviewer 2 Report

The review article by Dumarchey et al. describes mechanisms contributing to ineffective erythropoiesis in malaria. Information on various human and plasmodial factors have been nicely described. The paper is well-written and highlights key changes leading to the development of anemia in malaria, beyond increased splenic clearance and hemolysis. The authors have also used illustrations to describe the mechanisms discussed in the review. The paper provides novel information to scientists and physicians within the fields of tropical medicine, hematology and infectious diseases. As such, no specific weaknesses are noted in the review article.

Author Response

We are very grateful to the reviewer2 for his critical reading of the manuscript and his comments.